# Discovering User Types: Characterization of User Traits by Task-Specific Behaviors in Reinforcement Learning

## Abstract

We often want to infer user traits when personalizing interventions. Approaches like Inverse RL can learn traits formalized as parameters of a Markov Decision Process but are data intensive. Instead of inferring traits for individuals, we study the relationship between RL worlds and the set of user traits. We argue that understanding the breakdown of "user types" within a world – broad sets of traits that result in the same behavior – helps rapidly personalize interventions. We show that seemingly different RL worlds admit the same set of user types and formalize this observation as an equivalence relation defined on worlds. We show that these equivalence classes capture many different worlds. We argue that the richness of these classes allows us to transfer insights on intervention design between toy and real worlds.

## 1. Introduction

Mobile Health (mHealth) applications are becoming popular as cost-effective methods for improving health. The rapid personalization of treatment on which these apps depend is often achieved through Reinforcement Learning (RL). For example, on a physical therapy (PT) app, RL may be used to recommend daily exercises to a user who wants to regain ankle mobility. Effective personalization in RL often requires inferring information about users that is impossible to obtain directly. For example, a user's confidence in their physical capabilities and ability to perform long-term planning (their degree of myopia) significantly impact their success in prescribed fitness regimens. In our PT example, a confident user can adhere to challenging exercises. In contrast, a less-confident user may require simpler exercises to build a sense of mastery (Picha et al., 2021b). However, user traits such as confidence are difficult to infer, unlike

age, which is easy to query.

Formalizing the user as a Markov Decision Process (MDP), unknown user traits can be inferred using Inverse Reinforcement Learning (IRL), a process for learning MDP parameters (e.g. user's discount factor) by observing user behavior. However, standard IRL approaches are data-intensive (Yu et al., 2019), which poses a special challenge in mHealth settings because data collection from human experiments is costly. In our settings, we want to make informed design decisions about an intervention without extensive interactions with the user. We do so by reasoning a priori about the properties of the task and the set of possible user behaviors.

In this work, we formalize mHealth users as RL agents, and their traits, specifically their level of myopia and confidence, as parameters of a corresponding MDP (Section 2).

We argue that studying the breakdown of "user types" within a task – regions of the parameter space that define the same user behavior (i.e. optimal policy) – can inform the design and personalization of interventions, even before observing data. We visualize these user types in two-dimensional *behavior maps* (Section 3.1).

We demonstrate that seemingly different tasks admit the same behavior maps. We formalize this observation as an equivalence relation defined on tasks (Section 3.2). We demonstrate that each equivalence class, under our definition, is rich: we map several tasks commonly used in the RL literature to a few classes (Section 5.1).

Finally, we argue for the value of our equivalence definition applied to intervention design by providing an initial set of guidelines to generate insights for intervention design in real-life scenarios by lifting them from an equivalent, simpler toy world (Section 5.3).

**Related work.** We define an equivalence relation under which the set of user behaviors are similar amongst equivalent worlds, given a set of user traits. This mapping allows us to transfer interventions across many mHealth scenarios.

Many existing notions of equivalence in RL literature allow for knowledge transfer (Soni & Singh, 2006; Sorg & Singh, 2009). For example, equivalence definitions based

[1]Anonymous Institution, Anonymous City, Anonymous Region, Anonymous Country. Correspondence to: Anonymous Author <anon.email@domain.com>.

Preliminary work. Under review by the International Conference on Machine Learning (ICML). Do not distribute.

on bisimulation are used in MDP minimization, where one reduces large state spaces to speed up planning (Givan et al., 2003). Relaxed versions of bisimulations, for example, MDP homomorphism (Biza & Platt, 2018), stochastic homomorphism (van der Pol et al., 2020), and approximate homomorphisms (Ravindran & Barto, 2004) allow optimal policies in simple MDPs to be lifted to desirable policies in more complex, comparable MDPs. More general definitions of MDP equivalence can be defined through other methods of state aggregation (e.g. value-equivalence) (Li et al., 2006). Overall, these notions of equivalence are defined over the set of MDPs. In contrast, we decompose an MDP into task-specific and user-specific components, and we consider equivalences between the task-specific components of MDPs while allowing the user-specific components to vary (capturing different combinations of user traits).

Notions of equivalence from IRL are more directly related to our motivation of modeling user behavior, but they operate on a fundamentally different paradigm. IRL equivalence finds different rewards (Ziebart, 2010) or transitions (Reddy et al., 2018; Golub et al., 2013) that are equally likely under the demonstration data provided by *one* user. In contrast, our work assumes the existence of *multiple users*, and we want our equivalence to define which MDPs (worlds) will result in the same partition over these users.

Instead of learning the user's reward or transitions separately, Herman et al. (2016); Evans et al. (2016); Shah et al. (2019); Zhi-Xuan et al. (2020) learn them simultaneously. Like us, a subset of these works (Evans et al., 2016; Shah et al., 2019; Zhi-Xuan et al., 2020) model humans as RL agents with maladaptive MDPs. However, inferring an agent's (maladaptive) MDP parameters from demonstration is known to be a difficult and non-identifiable problem (Shah et al., 2019). We address this problem by defining an equivalence that allows us to anticipate which users will be indistinguishable in a given world.

## 2. Formalizing Users as RL Agents

We use Markov Decision Processes (MDP) to model human behavior in sequential decision-making tasks, such as an agent deciding to perform or not perform prescribed PT exercises. We refer to such tasks as *worlds*. User behavior within a world varies according to two traits, (1) the user's level of myopia and (2) the user's confidence level, which we capture as parameters of the MDP. For simplicity, in this paper, we only consider discrete state spaces.

**Worlds.** We define a world as a tuple $\mathcal{W} = \langle \mathcal{S}, \mathcal{A}, \mathcal{R} \rangle$ of states $\mathcal{S}$, actions $\mathcal{A}$, and rewards $\mathcal{R}$. This tuple captures the environment and the task in an application of interest (see Figure 2 for examples of grid worlds).

**The User MDP.** We define *the user MDP* as a tuple $\mathcal{M} = \langle \mathcal{S}, \mathcal{A}, \mathcal{T}_p, \mathcal{R}, \gamma \rangle$ of states $\mathcal{S}$, actions $\mathcal{A}$, transitions $\mathcal{T}_p$ parameterized by a user "confidence level" $p$, rewards $\mathcal{R}$, and discount factor $\gamma$. Here, $\gamma$ models the user's level of myopia, and $\mathcal{T}_p$ models the user's perceived world dynamics as a function of their confidence level $p$.

We split the user MDP into two components: user-specific parameters $\gamma$ and $p$ (the *user traits*), and application-specific world $\mathcal{W}$: $\mathcal{M} = \underbrace{\mathcal{W}}_{\text{application specific}} + \underbrace{\langle \gamma, T_p \rangle}_{\text{user specific}}$ .

An RL agent acts in the world, $\mathcal{W} = \langle \mathcal{S}, \mathcal{A}, \mathcal{R} \rangle$, according to policy $\pi : \mathcal{S} \rightarrow \mathcal{A}$, yielding a cumulative reward (expected returns): $J^\pi = \mathbb{E}\left[ \sum_{t=0}^{T} \gamma^t r_t \right]$. The optimal policy for the user MDP maximizes the expected returns: $\pi^* = \max_\pi J^\pi$. We assume that users are RL agents that follow the optimal policy for their corresponding user MDPs.

**User Behaviors.** Using our formalism, we can now model how and why users with different traits behave differently in the same world. For example, two people with different levels of myopia would judge different PT behaviors to be optimal in their respective MDPs. Our model attributes systematic differences in users' behavior (i.e. differences in their optimal policy) to differences in user traits (i.e. the user-specific parameters of their MDPs). In the following, we connect our formalization of user traits (their level of myopia, $\gamma$, and their confidence level, $p$) to well-studied constructs in psychology and behavioral science.

*Myopia* corresponds to the concept of *temporal discounting* in psychology. In user MDPs, we represent temporal discounting with $\gamma \in [0, 1)$. This captures people's tendency to undervalue future rewards, often leading to unhealthy behavior (Story et al., 2014). However, we note that in RL, discounting is exponential by default, which does not capture hyperbolic discounting (preference switching) observed in humans (Ainslie & Haslam, 1992; Shah et al., 2019).

In behavioral science, *confidence*, also known as self-efficacy, measures an agent's belief in their capability to perform a task (Picha et al., 2021a). Intuitively, this is the user's perceived probability that their intended outcome occurs due to their action. In user MDPs, we represent the user's confidence level with $p \in [0, 1]$, which is part of the transitions $\mathcal{T}_p$. Suppose $s'$ is the user's intended outcome for action $a$ from state $s$. Then, $T(s, a, s') = p$. We distribute the remaining $1 - p$ probability equally among the $\hat{s}' \in \hat{\mathcal{S}}$ possible outcomes: $T(s, a, \hat{s}') = \frac{1-p}{|\hat{\mathcal{S}}|}$. This formalization of confidence captures the phenomenon observed in psychology, where people with a higher belief in their own abilities persist for longer in the face of obstacles and adverse experiences (Bandura, 1977).

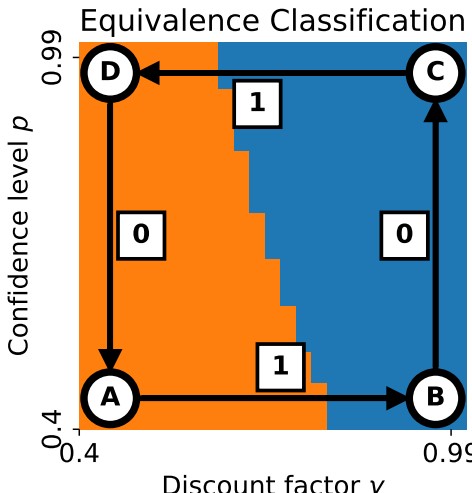

Figure 1: Example behavior map (Big-Small world). Annotations describe the procedure for deriving the equivalence class. The x-axis varies over discounting, $\gamma$; the y-axis varies over the confidence level, $p$. "Extreme" users, i.e. corners of the map, are labeled as circles. The number of "behavior switches" when tracing each edge between extreme users (from A to B, to C, to D, and back to A) are labeled as squares.

## 3. Equivalence Relation on the Set of Worlds

In this section, we define an equivalence relation on the set of worlds, identifying worlds that admit the same partition of the user parameter space (2-D space of $\gamma, p$) induced by user behaviors. This definition allows us to reason about one (usually more complex) world by transferring knowledge about user behaviors from an equivalent (simpler) world.

### 3.1. Behavior Maps

Given a world $\mathcal{W}$, we define a "behavior map" as the partitioning of the user parameter space, $\gamma \in [0, 1) \times p \in [0, 1]$, by user behavior (see Figure 1). We argue that behavior maps, denoted as $\mathcal{B}_\mathcal{W}$, can inform the design and deployment of interventions on user traits (for example, interventions to increase $\gamma$). Specifically, they can help us (1) determine to what extent user traits are identifiable through behavioral observations; (2) warm-start an intervention strategy for interacting with new users; and (3) estimate the required effect sizes for these interventions.

*Identifiability of User Traits.* Behavior maps allow us to anticipate the limits of what we can infer about a user using IRL or related methods, i.e. by observing their behavior in a given world. In fig. 1, we can distinguish between users with low and high discount factors by their corresponding behaviors but generally cannot distinguish between users

with different confidence levels. Thus, we would need information outside of what is typically collected for RL to infer confidence level in this setting.

*Warm-start Intervention Strategy.* Given a world and a new user, behavior maps can help select an intervention that, a priori, is likely to have more impact. In particular, the more variation in user behavior along a given axis, the more likely an intervention on the corresponding trait will change the user's behavior. For example, in fig. 1, we know that an intervention on $\gamma$ is more likely to change the user's behavior than an intervention on $p$.

*Intervention Effect Size.* If the behavior map has coarse partitions, effect sizes must be large, and variance in the effect must be tolerable. If the behavior map is finely partitioned, a small effect size may still accomplish behavior a desirable change. Still, a high variance in the effect could lead to unexpected and potentially undesirable changes.

Although useful, directly computing the behavior map for a complex application such as PT requires solving user MDPs for a range of user parameters – this is generally computationally impractical. Instead, to get the same insights, we reduce the PT world $\mathcal{W}$ to a simpler toy world $\mathcal{W}'$, for which we can easily compute $\mathcal{B}_{\mathcal{W}'}$. Below, we define an equivalence relation that allows us to make this reduction.

### 3.2. Equivalence Relation

*We consider two worlds similar if their corresponding behavior maps are equivalent.* Suppose two different applications, such as PT and dieting, have the behavior map from fig. 1: in both applications, confidence does not impact user behavior, and users with low gamma have one behavior, while users with high gamma have another. In this way, PT and dieting are similar worlds because insights that inform intervention design can be transferred from one to the other. The initial intervention strategy should focus on $\gamma$ instead of $p$ in both cases.

*We consider two behavior maps to be equivalent if the shapes of the decision boundaries between behaviors on the behavior map are the same.* We want our equivalence definition to be invariant to stretching or translating these boundaries. In our example, we want to say that PT and dieting have equivalent behavior maps if users with low $\gamma$ act differently from users with high $\gamma$. However, what is considered to be "low" or "high" $\gamma$ value need not match exactly between the two applications; hypothetically, in PT, the range for "low" $\gamma$ could be $[0, 0.3]$ and in dieting, the range could be $[0, 0.2]$.

In the following, we provide the formalism for equating two behavior maps. Intuitively, we consider each map as a diagram wherein $n_i$ number of vertices is placed on the $i$-th edge (indexing counter-clockwise from the lower left corner) and where each pair of vertices is connected by a

curve defined by a decision boundary separating two user behaviors. We say two maps are equivalent if, as diagrams, they are topologically equivalent under some transformation that preserves the ordering of the vertices on each edge. We now present this formally.

**Definition 3.1** (World Equivalence Induced by Behavior Map). We define an equivalence relation, $\equiv_{\mathrm{map}}$, on the set of discrete worlds $\mathfrak{W}$ by

$$\mathcal{W} \equiv_{\mathrm{map}} \mathcal{W}', \quad \mathcal{W}, \mathcal{W}' \in \mathfrak{W}$$

if and only if there is an ambient isotopy $h : \mathcal{B}_{\mathcal{W}} \times [0,1] \to \mathcal{B}_{\mathcal{W}'}$ between the decision boundaries in $\mathcal{B}_{\mathcal{W}}$ and the decision boundaries in $\mathcal{B}_{\mathcal{W}'}$. Furthermore, each $h_t$ is order-preserving when restricted to segments of the behavior map boundaries. That is, each

$$h_t|_{\{\gamma=0\}\times\{p\}}, h_t|_{\{\gamma=1\}\times\{p\}},$$
$$h_t|_{\{\gamma\}\times\{p=0\}}, h_t|_{\{\gamma\}\times\{p=1\}} : [0,1] \to [0,1],$$

is an order-preserving map.

The definition considers behavior maps in their entirety. However, we are specifically interested in worlds whose behavior maps are equivalent on the edges – that is, worlds for which the set of behaviors at extreme values of user parameters are identical.

In practice, computing the entire behavior map for arbitrarily complex applications can be impractical. Instead of relying on the entire behavior map, we simplify our equivalence definition to rely only on behavior at the extremes (edges of the map). Behavioral science can justify this simplification, as behaviors at the extremes are more extensively studied in psychology than average behaviors (e.g., the relationship between extreme discounting and health (Story et al., 2014)). In particular, this means that we may be able to use domain knowledge to quickly map a complex world to an equivalence class rather than directly solving a set of user MDPs.

**Definition 3.2** (World Equivalence Induced by Behavior Map Boundaries). We define an equivalence relation, $\equiv_{\partial\mathrm{map}}$, on the set of discrete worlds $\mathfrak{W}$ by

$$\mathcal{W} \equiv_{\partial\mathrm{map}} \mathcal{W}', \quad \mathcal{W}, \mathcal{W}' \in \mathfrak{W}$$

if and only if there is a ambient isotopy $h : \partial\mathcal{B}_{\mathcal{W}} \times [0,1] \to \partial\mathcal{B}_{\mathcal{W}'}$ between the decision boundaries in $\partial\mathcal{B}_{\mathcal{W}}$ and between the decision boundaries in $\partial\mathcal{B}_{\mathcal{W}'}$. Furthermore, each

$$h_t|_{\{\gamma=0\}\times\{p\}}, h_t|_{\{\gamma=1\}\times\{p\}},$$
$$h_t|_{\{\gamma\}\times\{p=0\}}, h_t|_{\{\gamma\}\times\{p=1\}} : [0,1] \to [0,1],$$

is an order-preserving map.

We note that applying Definition 3.2 reduces to counting the number of decision boundary components (i.e. behavior changes) along each edge of a behavior map (as described in Figure 1). That is, maps $M$ and $M'$ are equivalent under Definition 3.2 if the number of behavior changes on each map edge $M$ is equal to the number of behavior changes on the corresponding edge of map $M'$. Thus, indexing the edges, starting from the lower left corner, in a counter-clockwise fashion, each equivalence class can be represented simply as a vector $[n_1, n_2, n_3, n_4]$, where $n_i$ is the number of behavior changes along the $i$-th edge.

## 4. Atomic Worlds: Simple Representatives of Equivalence Classes

Under Definition 3.2, we seek the simplest representative worlds, called *atomic worlds*, for each equivalence class. User behaviors can be easily characterized in these toy worlds and the insights transfer to more complex worlds in the same equivalence classes. In our experiments, we identified three atomic worlds that other worlds commonly reduced to: Big-Small world, Cliff world, and Wall world. The worlds are visualized in Figure 2, and the corresponding behavior maps are given in Figure 3. The same real-world application can be mapped to different atomic worlds, depending on which aspects of the application we focus on. Using our PT application, we show how different aspects of user decision-making – choosing consistency of exercise, level of intensity, and type of therapy – map to different equivalent classes, represented by their atomic worlds – Big-Small, Cliff, and Wall, respectively.

The Big-Small world is characterized by a trade-off between choosing a smaller, more convenient reward and a bigger reward that is more difficult to reach. In mHealth, this world reflects scenarios in which smaller rewards, such as the time benefit of skipping PT for the day, preclude larger but delayed rewards, such as a fully rehabilitated ankle.

The Cliff world is characterized by a harmful absorbing state that may happen due to an action going awry. For example, deciding the intensity of the PT regimen can be modeled as a Cliff world. A high-intensity regimen could accelerate recovery but also risk re-injuring the patient.

The Wall world is characterized by choosing between a costly, short path to the goal and a longer, free path to the same goal. This can model the trade-off in choosing the type of physical therapy: virtual therapy may be more affordable, while in-person therapy is more costly and targeted.

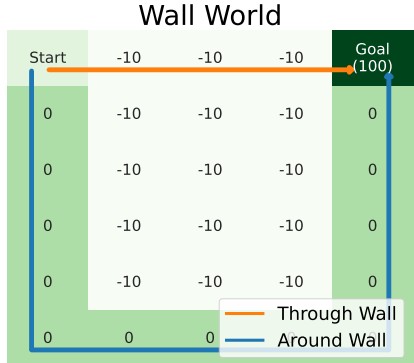

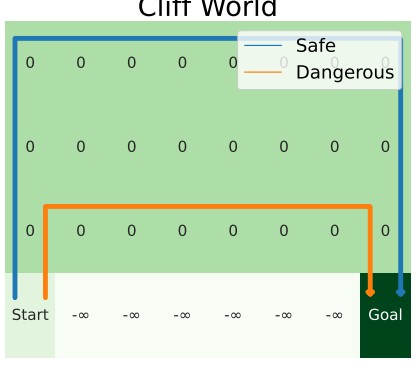

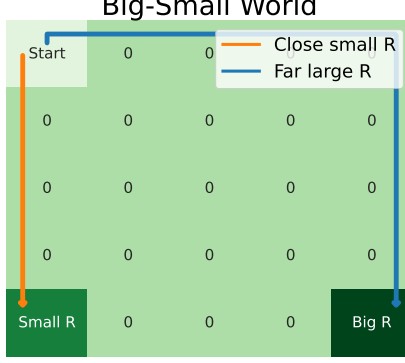

(a) A $6 \times 5$ Wall World where agents can pass directly through a costly wall (orange) or take the longer, safer path around it (blue).

(b) A $4 \times 8$ Cliff World where agents can walk close to the cliff and risk ruin (blue) or keep space but walk farther (orange).

(c) A $5 \times 5$ Big-Small World where agents can walk straight down to a small reward (orange) or farther to a bigger reward (blue).

Figure 2: Each atomic world has two qualitatively distinct behaviors (shown with blue and orange arrows). Each diagram displays what the world looks like. It also shows the possible behaviors within each world.

## 5. Empirical Analysis

### 5.1. Our equivalence classes are rich

We illustrate that our equivalence classes are rich by reducing multiple distinct worlds to one of our atomic worlds. As a result, we expect many worlds that model real-life applications can be reduced to these atomic worlds (or straightforward combinations of atomic worlds), allowing us to transfer our understanding of these simpler settings onto unexplored and seemingly complex ones.

We study several worlds commonly used in RL literature: Chain, RiverSwim, and Gambler's Fallacy (details in Appendix A). Under our definition, Chain (fig. 3d), RiverSwim (fig. 3e), and Gambler's Fallacy V1 (Figure 3f) are equivalent to Big-Small (Figure 3c); these are worlds in which the user chooses between a readily available but small reward (i.e., disengaging in Chain, swimming downstream in RiverSwim and Finishing in Gambler's) and a greater but more time-consuming reward. Gambler's Fallacy V2 (Figure 3g) is equivalent to Cliff World – both worlds have a "catastrophic absorbing state", that is, a nonzero risk of ending up in an absorbing state with a negative reward.

### 5.2. Our equivalence definition is robust to parameter perturbations in world definitions

We want a world to remain within its equivalent class despite minor parameter adjustments (e.g. the world for a month-long PT program should be in the same class as that for a 2-month program). This is evidence that our equivalence definition captures essential rather than incidental qualities of applications.

Concretely, we expect Big-Small to remain within its equivalence class despite parameter changes, such as the world's width or the ratio of the big to a small reward. We verify this in Figure 4. In Appendix B, we provide additional evidence of how our equivalence classes withstand perturbations across more parameters for all 7 worlds investigated.

### 5.3. Our equivalence definition can inform real-world intervention design

In this section, we provide general guidelines for how one would use our equivalence definition to generate intervention insights for real-world applications:

1. **Mapping real world to atomic world:** Many real-world applications can be quickly mapped to an atomic world with domain knowledge. For example, behavior scientists can often describe expected user behavior at the extremes – e.g. "how many different behaviors are there for users with very low confidence?"

2. **Lifting insights from atomic world to real world:** Given a fixed intervention, if the intervention depends on inferring $\gamma$ or $p$ more precisely than the atomic world indicates, an external source of information is needed (e.g. users take an additional questionnaire to estimate their discount factor).

   Given a choice of interventions, we can select an intervention based on the mean and variance of the effect size needed for the task, according to the behavior map of the atomic world. For example, if the application reduces to a Wall world (see fig. 3a), and it's desirable for users to go through the Wall, then the effect on $\gamma$ must not exceed $\approx 0.99 - 0.71 = 0.28$.

   We can also decide on an initial intervention strategy. If the atomic world indicates user behaviors vary more along one axis (e.g. user behaviors differ by $\gamma$ but not

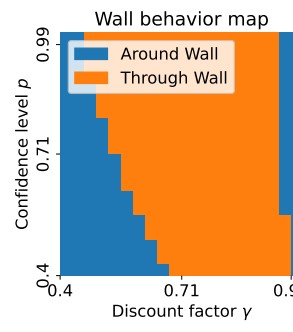

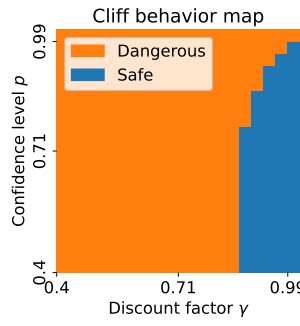

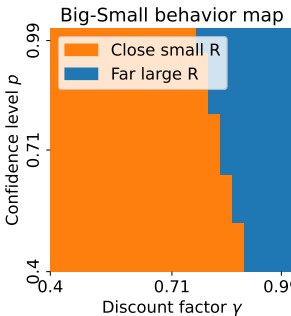

(a) Wall world example behavior map belonging to equivalence class [2, 0, 2, 0].

(b) Cliff world example behavior map belonging to equivalence class [1, 1, 0, 0].

(c) Big-Small world example behavior map belonging to equivalence class [1, 0, 1, 0].

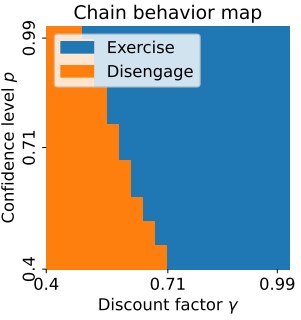

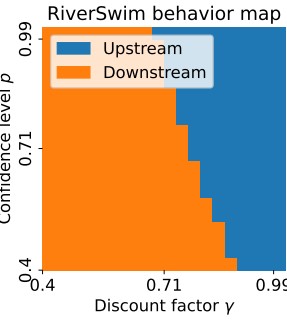

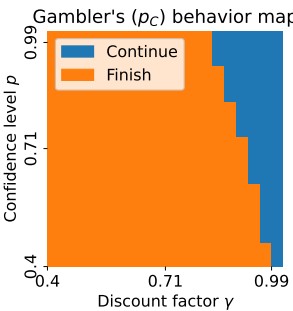

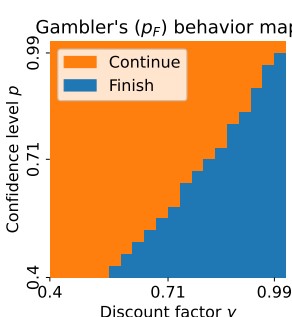

(d) Chain world example behavior map belonging to equivalence class [1, 0, 1, 0].

(e) RiverSwim world example behavior map belonging to equivalence class [1, 0, 1, 0].

(f) Gambler's Ruin world (with varying $p_C$) example behavior map belonging to equivalence class [1, 0, 1, 0].

(g) Big-Small world example behavior map belonging to equivalence class [1, 0, 1, 0].

Figure 3: This figure shows that seemingly different worlds (bottom row) produce behavior maps that can be classified into the same class as one of our atomic worlds (top row).

$p$), then the initial strategy is to intervene on $\gamma$.

since complicated scenarios can be broken down into atomic worlds that each capture a unique dynamic of the MDPs.

## 6. Discussion & Future Work

We expect there exist more atomic worlds than the three explored in this paper, especially when worlds have more than two qualitatively distinct strategies. Further work involves finding new classes and categorizing novel scenarios.

Furthermore, our RL-focused analysis of grid worlds would benefit from experiments on human behavior. For example, the equivalence classifications can be supported by studies of ways to map between real-life mHealth settings and atomic world MDPs, leveraging existing methods and knowledge from behavioral science and psychology.

Complex real-life scenarios will unlikely map neatly onto a singular atomic world; however, we conjecture that some worlds may fall into *compositions* of atomic worlds. Some initial experiments with composite worlds indicate that the Big-Small and Cliff world composition leads to a behavior map that combines the atomic worlds' respective maps. This further supports the generality of our equivalence classes

## 7. Conclusion

This work proposes a novel method for informing mHealth interventions that affect user behavior through changing user traits. We define an equivalence relation between sequential decision-making tasks based on how user behaviors partition the configuration space of user traits. We demonstrate that a number of seemingly different RL scenarios map to one of a few equivalence classes, each represented by a simple toy world. In particular, we argue that many real-world applications can be mapped to simple toy worlds by leveraging domain knowledge in behavioral science and psychology. We show how insight into simple intervention design can be lifted to complex settings within the same equivalence class. We believe that our work highlights an interesting direction for intervention design that complements or bypasses data-intensive inference of user traits via traditional IRL.

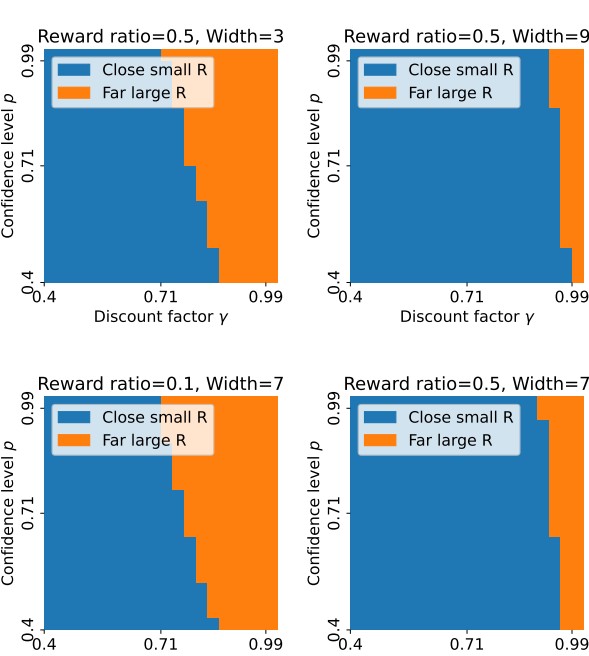

Figure 4: A Big-Small world stays within its equivalence class for many different parameter combinations. The example behavior maps have different values for the world width and the magnitude of the far-away large reward, while the rest of the parameters are fixed as `height = 7` and `Big far R = 300`.

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

# A. Descriptions of Each World from the Literature

Below, we present the MDPs for the MDPs from the mHealth literature we study in this work, i.e., the Chain World, RiverSwim World, and Gambler's Ruin in Figure 5, Figure 6, and Figure 7. Blue arrows indicate the behavior corresponding to the blue behavior in the corresponding behavior maps, likewise for orange arrows.

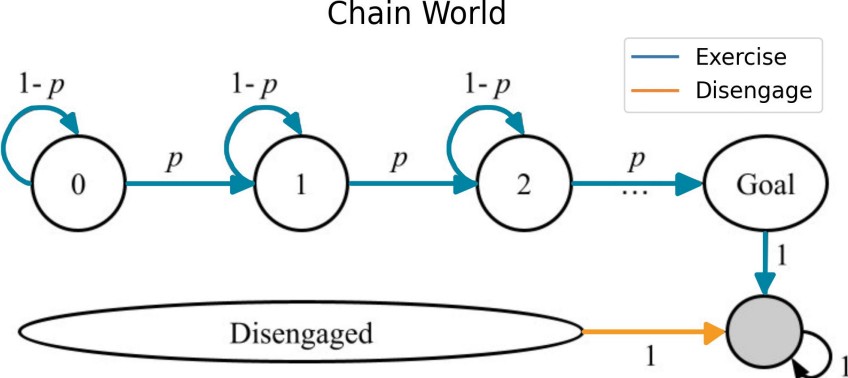

Figure 5: In the Chain world, users can choose to "exercise," or progress step-by-step to reach the desired goal. At each stage, they also have the option to "disengage," which results in a smaller reward and the termination of their progression.

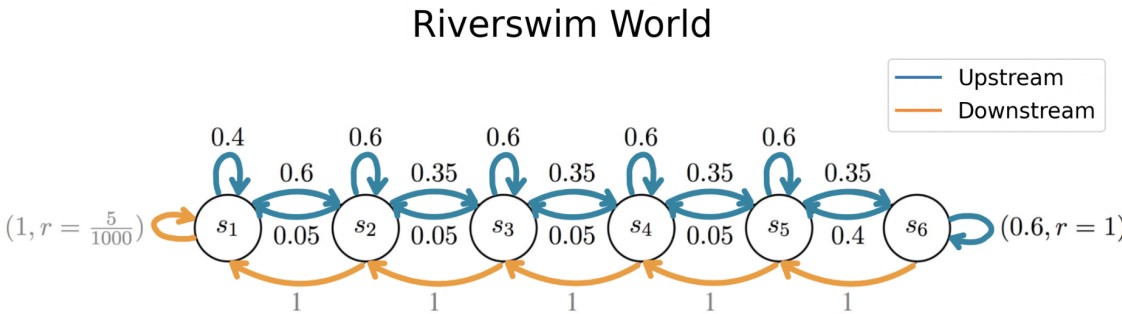

Figure 6: In the RiverSwim world, the user can choose the rightward "upstream" action, which has a chance of successfully advancing the user toward the larger reward but also a failure probability of staying in place or falling behind. They can also choose the leftward "downstream" action that deterministically moves the user toward the small reward on the far left.

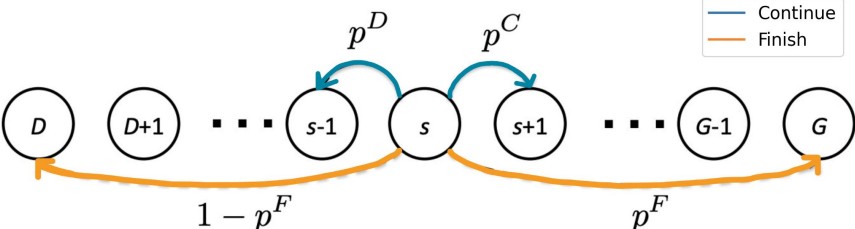

Figure 7: In the Gambler's Ruin (Bandit Problem) world, users can choose the "continue" action, which can either move the user one step left toward the dead-end state or one step right toward the goal state. They can also choose the "finish" action, moving them directly to the dead-end or goal state.

## B. Parameter Perturbations for Each World

Below, we present more comprehensive investigations into the invariance of the different worlds to changes in the world parameters under our definition of equivalence. Different worlds have different sets of parameters to perturb and different ranges for which they remain invariant.

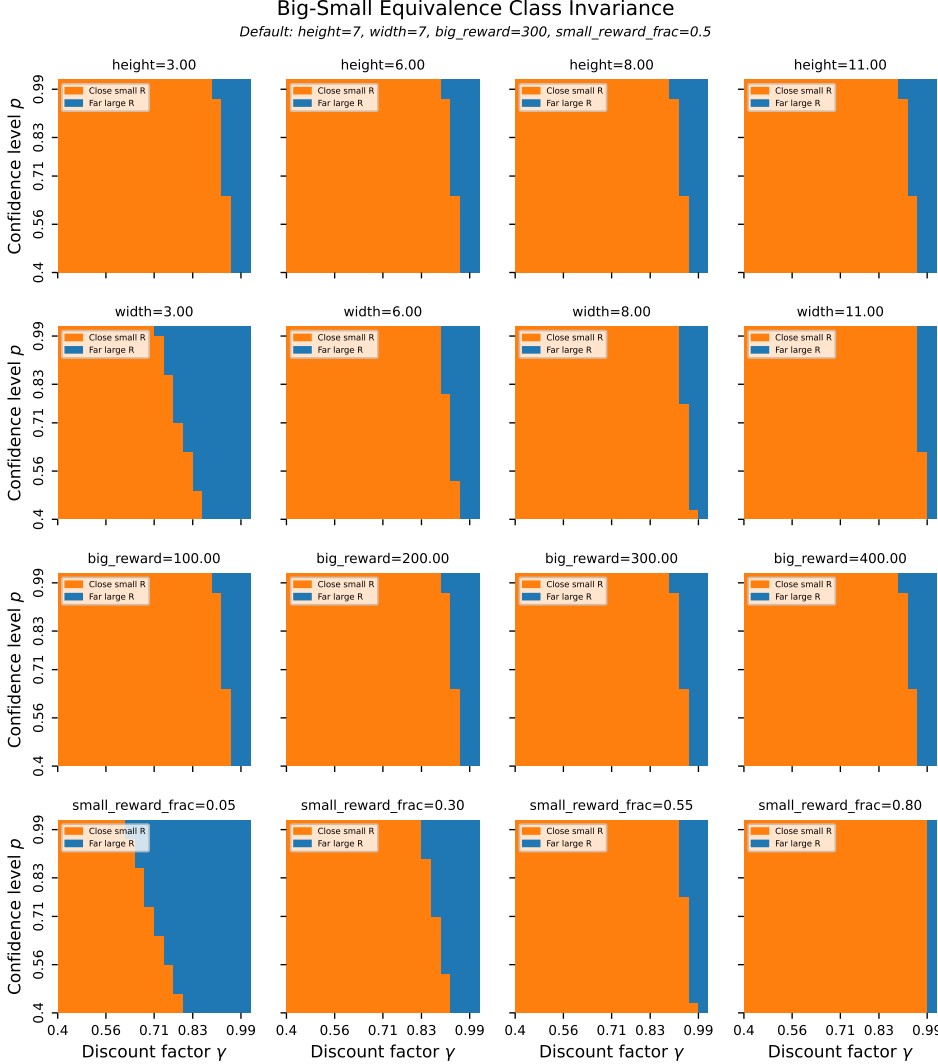

Figure 8: This array of graphs depicts behavior maps within the Big-Small world across variations of multiple parameters, such as world size and magnitude of rewards. While the graphs are not identical, all these maps are still in the same equivalence class under our definition, indicating their robustness to parameter perturbations.

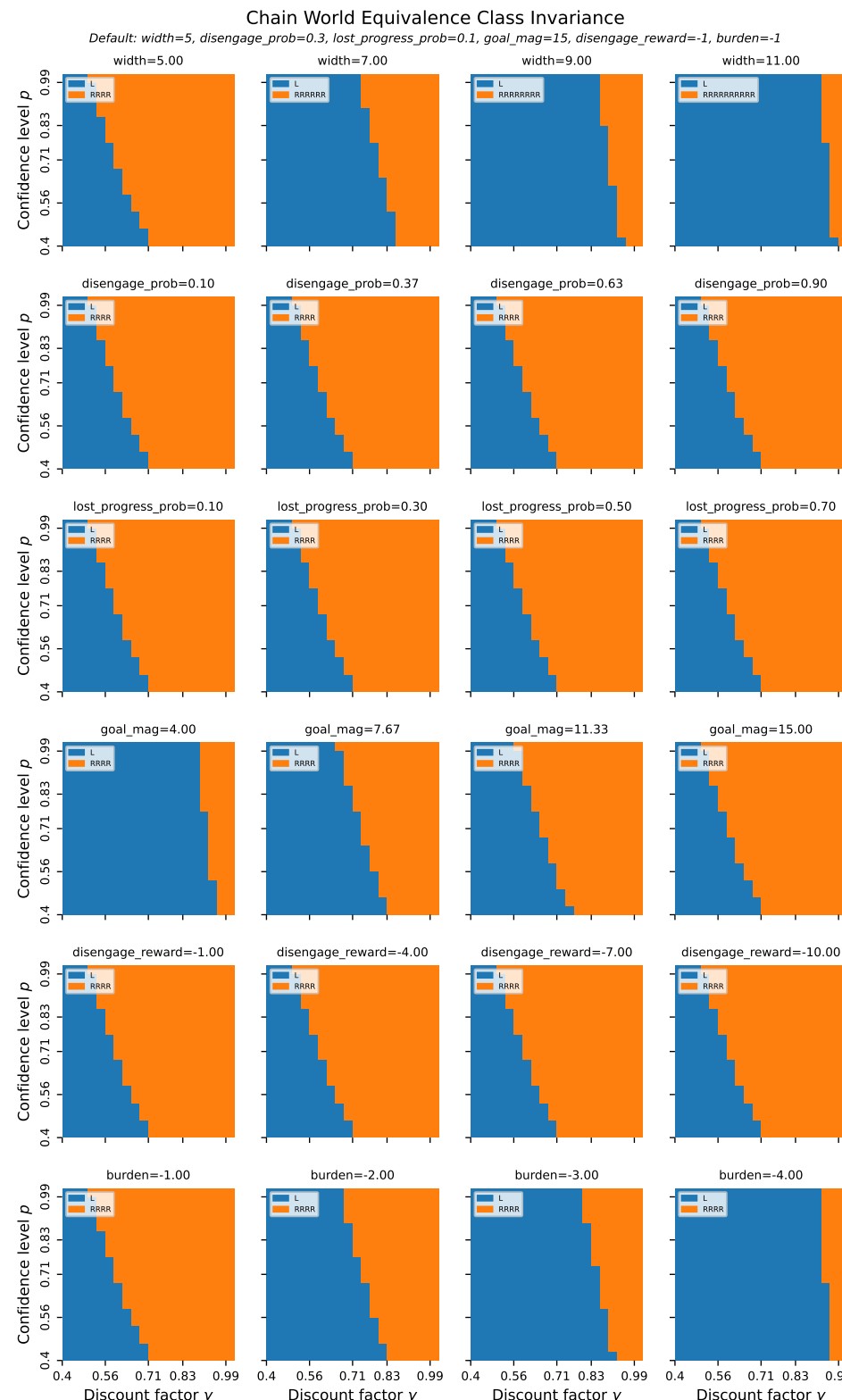

Figure 9: This array of graphs depicts behavior maps within the Chain world across variations of six parameters, including world size and disengagement probabilities. These maps are placed in the same equivalence class under our definition, indicating their robustness to parameter perturbations.

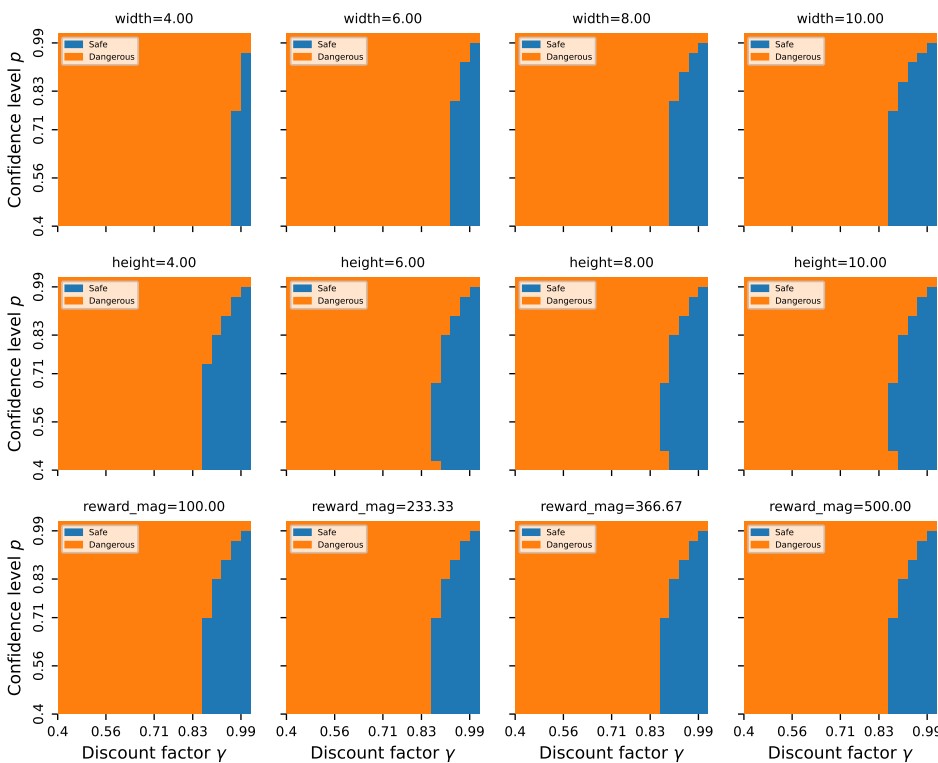

Figure 10: This array of graphs depicts behavior maps within the Cliff world across variations of three parameters: height, width, and reward size. These maps are placed in the same equivalence class under our definition, indicating their robustness to parameter perturbations.

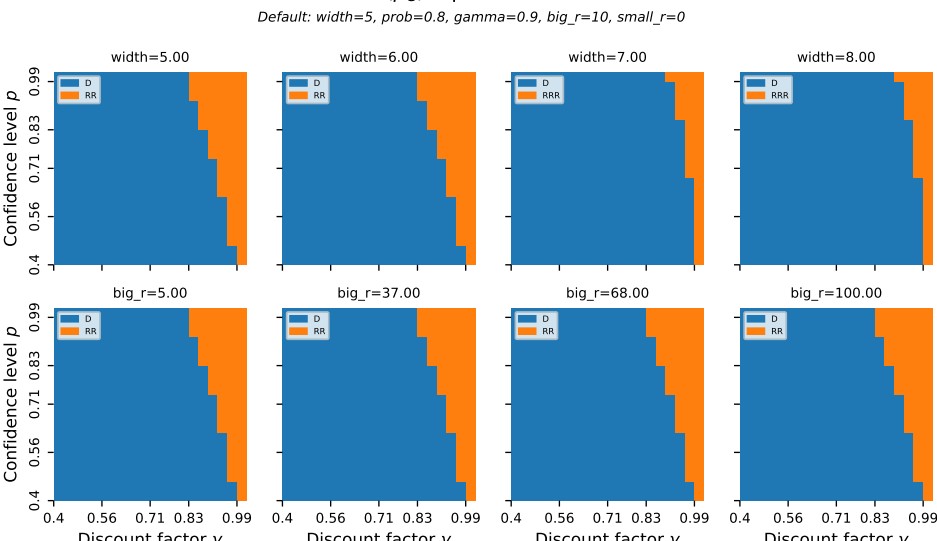

Figure 11: This array of graphs depicts behavior maps within the Gambler's Ruin world across the width and reward size variations while holding the failure probability $(p^F)$ constant. These maps are placed in the same equivalence class under our definition, indicating their robustness to parameter perturbations.

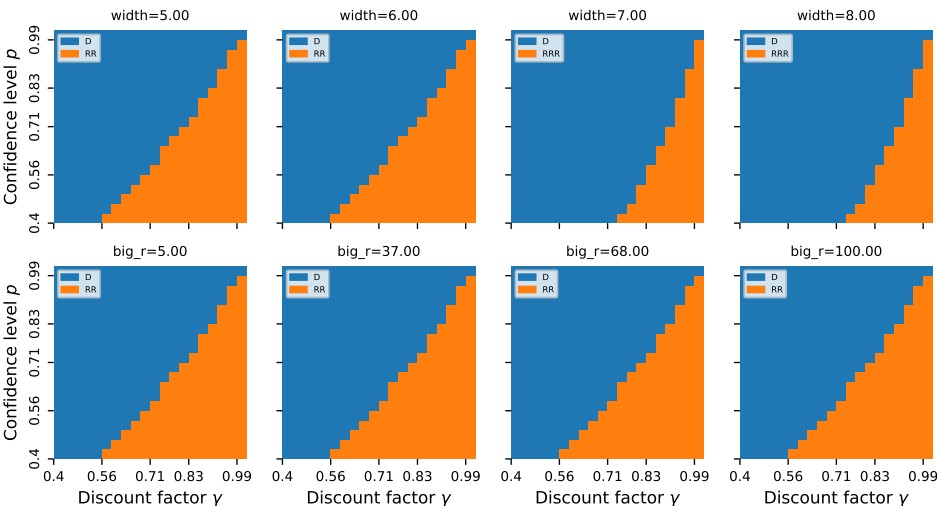

Figure 12: This array of graphs depicts behavior maps within the Gambler's Ruin world across the width and reward size variations while holding the "continue" probability $\left(p^C\right)$ constant. These maps are placed in the same equivalence class under our definition, indicating their robustness to parameter perturbations.

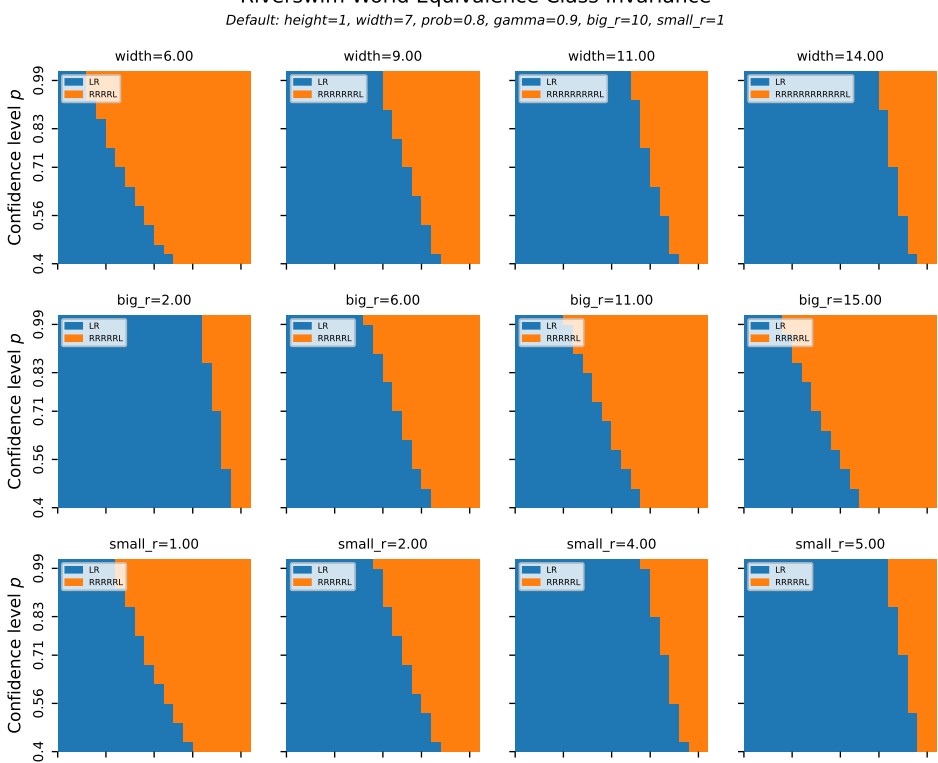

Figure 13: This array of graphs depicts behavior maps within the RiverSwim world across the width and reward sizes variations. These maps are placed in the same equivalence class under our definition, indicating their robustness to parameter perturbations.

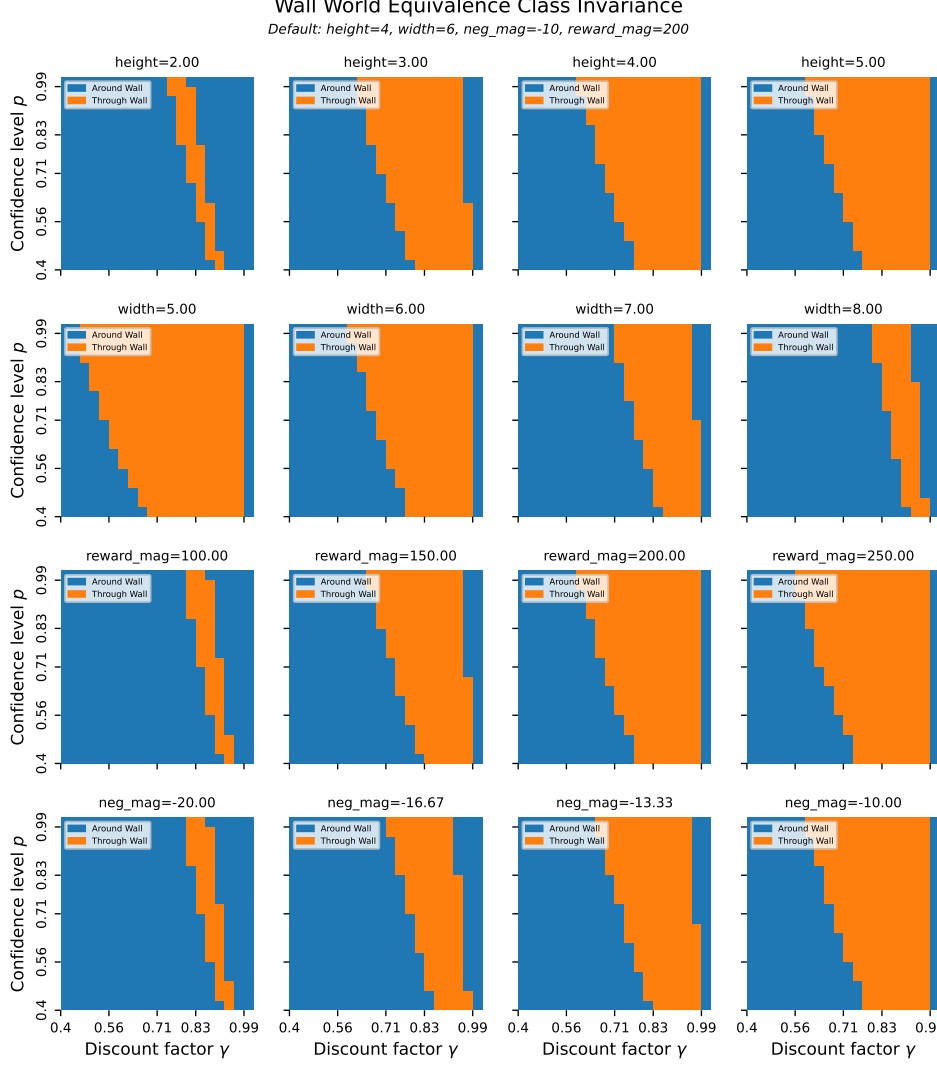

Figure 14: This array of graphs depicts behavior maps within the Wall world across variations of world size and reward magnitude. These maps are placed in the same equivalence class under our definition, indicating their robustness to parameter perturbations.

