# OpenReview forum: "Discovering User Types: Characterization of User Traits by Task-Specific Behaviors in Reinforcement Learning"
_ICML.cc/2023/Workshop/ILHF — ILHF Workshop ICML 2023_

### Official Review · Reviewer_QyYz · 2023-06-04

**Rating:** 5
**Confidence:** 3

**Review:**

This paper studies how to infer personalized user traits. The key idea is to model each user as an RL agent. Then the paper proposes a behavior map, a partitioning of the user parameter space, to model the user behaviors. Behavior maps allow us to anticipate the limits of what we can infer about a user using Inverse RL or related methods. Experiments on several Mini World environments show the effectiveness of the proposed analytic tools.

Strengths:
1. The idea is very novel and interesting.
2. Experiment results on several environments show effectiveness.

Weaknesses: The main weakness is that the tested environments are too simple. The experiments should be conducted in meaningful environments, such as recommendations, where understanding users' personalized behaviors is meaningful.

---

### Official Review · Reviewer_8huw · 2023-06-16
**Clear presentation and valid approach; still lacking steps to support the proposed real world applications**

**Rating:** 6
**Confidence:** 4

**Review:**

Summary:
The paper proposes characterizing user behaviors in MDP agents with different “discount factor” and “confidence”, which results in user behavior maps in grid-world MDPs. The author shows the generality of the framework by reducing three commonly studied grid world two the same behavior map (Big-Small).

Pros:
    1. The paper uses succinct language, and good visualizations to illustrate the point. The behavior maps and MDP visualizations are very easy to follow.

Cons:
    1. Overall, I agree with the authors that characterizing behaviors is more direct than characterizing user traits in certain applications (i.e. different traits can lead to the same behaviors), the authors may need to show that this is actually better than learning user traits. Thus, I recommend also comparing with IRL or equivalent algorithms.
    2. Section 3 proposes a number of definitions of equivalence in behavior map. However, it is unclear to me what these definitions lead to in terms of theoretical results, or difference in experiment design/empirical results. The author needs to either validate these definitions, or build on top of them.
    3. I left a bit confused about how interventions can be performed, in order to help improve patient results. While I understand that changing user gamma and T_p can lead to behavior changes, how do real-world interventions correlate with user changes in these parameters? It would be helpful if the authors could provide experiments (simulation or real world) or examples to illustrate this. Otherwise, the paper reads a bit detached from mHealth applications.

---

### Decision · Program_Chairs · 2023-06-20

Accept